# RANDOM NETWORK DISTILLATION AS A DIVERSITY METRIC FOR BOTH IMAGE AND TEXT GENERATION

## ABSTRACT

Generative models are increasingly able to produce remarkably high quality images and text. The community has developed numerous evaluation metrics for comparing generative models. However, these metrics do not effectively quantify data diversity. We develop a new diversity metric that can readily be applied to data, both synthetic and natural, of any type. Our method employs random network distillation, a technique introduced in reinforcement learning. We validate and deploy this metric on both images and text. We further explore diversity in few-shot image generation, a setting which was previously difficult to evaluate.

## 1 INTRODUCTION

State-of-the-art generative adversarial networks (GANs) are able to synthesize such high quality images that humans may have a difficult time distinguishing them from natural images (Brock et al., 2018; Karras et al., 2019). Not only can GANs produce pretty pictures, but they are also useful for applied tasks from projecting noisy images onto the natural image manifold to generating training data (Samangouei et al., 2018; Sixt et al., 2018; Bowles et al., 2018). Similarly, massive transformer models are capable of performing question-answering and translation (Brown et al., 2020). In order for GANs and text generators to be valuable, they must generate diverse data rather than memorizing a small number of samples. Diverse data should contain a wide variety of semantic content, and its distribution should not concentrate around a small subset of modes from the true image distribution.

A number of metrics have emerged for evaluating GAN-generated images and synthetic text. However, these metrics do not effectively quantify data diversity, and they work on a small number of specific benchmark tasks (Salimans et al., 2016; Heusel et al., 2017). Diversity metrics for synthetic text use only rudimentary tools and only measure similarity of phrases and vocabulary rather than semantic meaning (Zhu et al., 2018). Our novel contributions can be summarized as follows:

- We design a framework (RND) for comparing diversity of datasets using random network distillation. Our framework can be applied to any type of data, from images to text and beyond. RND does not suffer from common problems that have plagued evaluation of generative models, such as vulnerability to memorization, and it can even be used to evaluate the diversity of natural data (not synthetic) since it does not require a reference dataset.

- We validate the effectiveness of our method in a controlled setting by synthetically manipulating the diversity of GAN-generated images. We use the same truncation strategy employed by BigGAN to increase FID scores, and we confirm that this strategy indeed decreases diversity. This observation calls into question the usefulness of such popular metrics as FID scores for measuring diversity.

- We benchmark data, both synthetic and natural, using our random distillation method. In addition to evaluating the most popular ImageNet-trained generative models and popular language models, we evaluate GANs in the data scarce regime, i.e. single-image GANs, which were previously difficult to evaluate. We also evaluate the diversity of natural data.

## 2  DESIGNING A GOOD DIVERSITY METRIC

Formally defining "diversity" is a difficult problem; human perception is hard to understand and does not match standard mathematical norms. Thus, we first define desiderata for a useful diversity metric, and we explore the existing literature on evaluation of generative models.

### 2.1  WHAT DO WE WANT FROM A DIVERSITY METRIC?

**Diversity should increase as the data distribution's support includes more data.** For example, the distribution of images containing brown dogs should be considered less diverse than the distribution of images containing brown, black, or white dogs. While this property might seem to be a good stand-alone definition of diversity, we have not yet specified what types of additional data should increase diversity measurements.

**Diversity should reflect signal rather than noise.** If a metric is to agree with human perception of diversity, it must not be highly sensitive to noise. Humans looking at static on their television screen do not recognize that this noise is different than the last time they saw static on their screen, yet these two static noises are likely far apart with respect to $l_p$ metrics. The need to measure semantic signal rather than noise precludes using entropy-based measurements in image space without an effective perceptual similarity metric. Similarly, diversity metrics for text that rely on counting unique tokens may be sensitive to randomly exchanging words with their synonyms, or even random word swaps, without increasing the diversity of semantic content.

**Quality $\neq$ diversity.** While some GANs can consistently produce realistic images, we do not want to assign their images a high diversity measurement if they produce very little variety. In contrast, other GANs may produce a large variety of unrealistic images and should receive high diversity marks. The quality and diversity of data are not the same, and we want a measurement that disentangles the two.

**Metrics should be agnostic to training data.** Recent single-image GANs and few-shot GANs are able to generate many distinct images from very few training images (sometimes just one) (Shaham et al., 2019b; Clouâtre & Demers, 2019). Thus, a good metric should be capable of producing diversity scores for synthetic data that are higher than those of the training set. Likewise, simply memorizing the training data should not allow a generative model to achieve a maximal diversity score. Moreover, two companies may deploy face-generating models trained on two disjoint proprietary datasets, and we should still be able to compare the diversity of faces generated by these models without having training set access. An ideal diversity metric would allow one to collect data and measure its diversity outside of the setting of generative models.

**Diversity should be measureable on many kinds of data.** Measurements based on hand-crafted perceptual similarity metrics or high-performance neural networks trained carefully on large datasets can only be used for the single type of data for which they are designed. We develop a diversity concept that is adaptable to various domain, including both images and text.

### 2.2  EXISTING METRICS

We now review existing metrics for generative models to check if any already satisfy the above criteria. We focus on the most popular metrics before briefly discussing additional examples.

**Inception Score (IS) (Salimans et al., 2016).** The Inception Score is a popular metric that rewards having high confidence class labels for each generated example, according to an ImageNet trained InceptionV3 network, while also producing a diversity of softmax outputs across the overall set of generated images (Deng et al., 2009; Szegedy et al., 2016). While this metric does encourage generated data to be class-balanced and is not fooled by noise, IS suffers from several disqualifying problems when considered as a measure of diversity. First, it does not significantly reward diversity within classes; a generative model that memorizes one image from each class in ImageNet may achieve a very strong score. Second, IS often fails when used on classes not in ImageNet and is not adaptable to settings outside of natural image classification (Barratt & Sharma, 2018). Finally, IS does not disentangle diversity from quality. The Inception Score can provide a general evaluation of GANs trained on ImageNet, but it has limited utility in other settings.

**Fréchet Inception Distance (FID) (Heusel et al., 2017).** The FID score measures the Fréchet distance between a Gaussian distribution fit to InceptionV3 features on generated data and a Gaussian distribution fit to features on ground-truth data, for example natural images on which the generative model was trained. Unlike IS, FID scores compare generated data to real data, and they do not explicitly rely on ImageNet classes. Thus, FID more effectively discourages a generator from memorizing one image per class. However, FID assumes that the training data is "sufficient" and does not reward producing more diversity than the training data. A second problem with FID is that it relies on either (1) ImageNet models being useful for the problem at hand or (2) the ability to acquire a reference dataset and train a high-performance model on the problem. Like IS, FID does not disentangle diversity from quality and suffers from similar problems.

Several additional metrics have attempted to address these problems. For example, precision and recall have been suggested to tease apart different aspects of a generative model's performance (Sajjadi et al., 2018; Kynkäänniemi et al., 2019). However, these metrics (as used to measure diversity), along with other earlier metrics all aim to estimate the likelihood of real data under the generated distribution, and thus, require access to a reference ground-truth dataset. This makes these metrics are inflexible, and these metrics achieve optimal values when a generator simply memorizes the reference dataset. Other examples of metrics include the Modified Inception Score (m-IS) uses a cross-entropy term to resolve the fact that IS rewards models for memorizing one image per ImageNet class (Gurumurthy et al., 2017). However, m-IS is still beholden to the InceptionV3 label space and still prevents a user from discerning diversity from quality. Boundary distortion is a method for detecting covariate shift in GAN distributions by comparing how well classifiers trained on synthetic data perform on ground-truth data (Santurkar et al., 2018). This measurement indicates similarity to the true data distribution rather than purely measuring diversity, and it ignores the possibility of models generating even more diverse data than the data on which they are trained. Additionally, this method assumes the user has a large ground-truth dataset. Recently, classification accuracy score (CAS) was introduced to measure the performance of generative models by training classifiers on synthetic data produced by the generative models and evaluating the performance of the classifier (Ravuri & Vinyals, 2019). Similar to the boundary distortion method, this measurement focuses its comparison on the likeness of the synthetic data distribution to the ground-truth distribution and is dependent upon having access to labeled ground-truth data.

In natural language processing, several metrics exist for evaluating text generation for dialogue, conversational AI systems, and machine translation. Papineni et al. (2002) introduced the now widely used **Bilingual Evaluation Understudy (BLEU)** score as a metric for evaluating the quality of machine translated text. BLEU uses a modified form of precision to compare a candidate translation against multiple reference translations, where diversity ideally can be evaluated by including all plausible translations as references when computing the score. However, this requires massive annotation cost, and it remains difficult to capture all viable translations for a given sentence. Li et al. (2015) and Xu et al. (2018) propose counting the number of unique n-grams as a measure for evaluating the diversity of text generation tasks in conversational models, however, this metric does not account for the semantic meaning of different tokens and fails to capture paraphrases of semantically similar text. Montahaei et al. (2019) propose a joint metric for assessing both quality and diversity for text generation systems by approximating the distance of the learned generative model and the real data distribution. This metric couples both diversity and quality in a single metric. Zhu et al. (2018) introduce Self-BLEU to evaluate sentence variety. Self-BLEU measures BLEU score for each generated sentence by considering other generations as references. By averaging these BLEU scores, a metric called Self-BLEU is computed where lower values indicate more diversity. However, Self-BLEU remains very sensitive to local syntax, and it fails to capture global consistency and diverse semantic information in generated text.

Many other metrics exist for generative models which do not approach the problem of diversity. Our work is not the first to recognize this gap in the literature (Borji, 2019).

## 3 MEASURING DIVERSITY WITH RANDOM NETWORK DISTILLATION

### 3.1 RANDOM NETWORK DISTILLATION

Random Network Distillation (RND) was first introduced as an exploration bonus for RL agents (Burda et al., 2018). The bonus is tied to how well an agent could predict the random features ex-

tracted from its environment. This served as a useful proxy for how "novel" the agent's environment was. This bonus is designed to overcome a reward trap wherein RL agents gets "stuck" in an environment. Agents could, for example, get stuck in front of an ever-changing static TV screen because the agent is rewarded for seeing a "novel" environment - a new instance of random noise. With the RND bonus, the predictor network quickly learns to predict the features of the static TV screen, and the agent is not rewarded for remaining in the same environment.

## 3.2 OUR APPROACH

We propose a diversity measurement framework, the RND score, motivated by random network distillation. In our method, after randomly splitting the data into train and validation sets, we train a predictor network to predict the feature vectors output by a randomly initialized target network. This target network is never trained. Then, the RND score is the average (normalized) generalization gap at the end of training. Intuitively, when data is diverse, we expect training data to differ significantly from validation data, while when data is not diverse, we expect training to to be similar to testing data. Thus, a predictor network trained on training data from a very diverse data distribution should have a harder time predicting the output of the target network on validation data. The following framework can be applied to any type of data by choosing a network architecture, size of training split, and a training routine appropriate for the setting. In Appendix A.4, we demonstrate that RND-based comparisons between datasets are stable under these choices.

Given a dataset $S$, a target network $\mathcal{T}$, and a predictor network $\mathcal{P}$, both with architecture $\mathcal{A}$, let

$$\mathrm{MSE}(\mathcal{T}, \mathcal{P}; S) = \frac{1}{|S|} \sum_{x \in S} \|\mathcal{T}(x) - \mathcal{P}(x)\|_2^2. \tag{1}$$

We randomly initialize target and predictor networks, $\mathcal{T}$ and $\mathcal{P}$, prior to training, and we randomly split dataset $S$ into train/validation splits, $S_t$ and $S_v$. After $i$ epochs of training the predictor network on this mean squared error loss, let

$$\mathrm{RND}_i(S) = \frac{\mathrm{MSE}(\mathcal{T}, \mathcal{P}_i; S_v) - \mathrm{MSE}(\mathcal{T}, \mathcal{P}_i; S_t)}{\mathrm{MSE}(\mathcal{T}, \mathcal{P}_i; S_v) + \mathrm{MSE}(\mathcal{T}, \mathcal{P}_i; S_t)}. \tag{2}$$

Then, the RND score is given by

$$\mathrm{RND}(S) = \mathbb{E}\left[\frac{1}{n - n_0 + 1} \sum_{i=n_0}^{n} \mathrm{RND}_i(S)\right], \tag{3}$$

where $n_0$ and $n$ denote start and end epochs for measuring $\mathrm{RND}_i$, respectively, and the expectation is taken over random data splits and network initializations. We now discuss several design choices.

**Why random networks?** We leverage randomly initialized feature extractors since this keeps the RND score independent of any auxiliary datasets. Large, pre-trained feature extractors, as seen in the calculation of FID, are trained to extract useful features for classification problems on a fixed data distribution. While these feature extractors are often able to extract salient features from other, like distributions, the utility of pre-trained feature extractors diminishes on drastically different distributions. Furthermore, we want to separate image fidelity from image diversity. Scores like FID, which rely upon pre-trained feature extractors, are sensitive to changes in image quality (Ravuri & Vinyals, 2019). Random features are well-studied, both for kernel methods and neural networks (Rahimi & Recht, 2008; Rudi & Rosasco, 2017; Yehudai & Shamir, 2019; Mei & Montanari, 2019).

**Why normalize the generalization gap?** We normalize the generalization gap by the magnitudes of its components in order to promote scale invariance. Otherwise, data whose entries have larger magnitudes may receive a higher diversity score, despite having similar semantic content.

Consider the simple case where the target and trained predictor networks, $\mathcal{T}$ and $\mathcal{P}$, are linear, and consider a dataset split into $S_t$, $S_v$. Now, consider the datasets, $2S_t$ and $2S_v$, formed by multiplying each element of the original datasets by 2. Since both learning problems are convex, and the target and predictor networks are of the same family, $\mathcal{P} = \mathcal{T}$ and $\mathrm{MSE}(\mathcal{T}, \mathcal{P}; 2S_t) = \mathrm{MSE}(\mathcal{T}, \mathcal{P}; S_t) = 0$. However, the generalization gaps have the property that $\mathrm{MSE}(\mathcal{T}, \mathcal{P}; 2S_v) - \mathrm{MSE}(\mathcal{T}, \mathcal{P}; 2S_t) = 2^2[\mathrm{MSE}(\mathcal{T}, \mathcal{P}; S_v) - \mathrm{MSE}(\mathcal{T}, \mathcal{P}; S_t)]$ since $\mathcal{T}(2x) - \mathcal{P}(2x) = 2(\mathcal{T}(x) - \mathcal{P}(x))$. In other words, the generalization gap increases substantially if we do not normalize.

**How do we estimate the RND score?** We estimate the expected value in equation 3 by training the predictor network with multiple data splits. On each run, we average $RND_i$ at several epochs, $n_0$ through $n$, to remove noise at no extra computational cost. Experiments concerning the stability of this estimate can be found in Appendix A.4. See Appendix A.1 Algorithm 1 for a sketch of the RND score computation pipeline. The exact architecture and training procedure depends on the setting. For example, we use a transformer architecture to evaluate text, and we use a ResNet architecture to evaluate images (Vaswani et al., 2017; He et al., 2016). More details about training, and ablation studies, can be found in appendices A.3 and A.4.

## 4 EXPERIMENTS

### 4.1 VALIDATING RND IN A CONTROLLED SETTING

In order to validate RND in a controlled setting, we synthetically manipulate the diversity of GAN-generated images by truncating latent distributions. The "truncation trick" allows generative models, such as BigGAN, to improve image quality at the cost of diversity (Brock et al., 2018). Intuitively, the generator produces better looking (but less diverse) images given a latent vector coming from a truncated distribution, since similar latent vectors were more likely to be sampled during training. Thus, a good diversity metric should be able to measure the trade-off in diversity that occurs when the latent truncation parameter is tuned. It is worth noting that unlike RND, FID scores do not always rank images according to the extent of their latent truncation. Ravuri & Vinyals (2019) show that while FID scores do improve as the latent truncation parameter grows from a very small value, FID scores eventually degrade as the truncation parameter increases. This behaviour may occur because FID is also sensitive to image quality - a characteristic that makes it ill-suited to measure diversity.

We validate our diversity measurement by generating images from three different truncation parameters in the range found on the TensorFlow Hub implementation of BigGAN. We see in Figure 1 that the RND scores perfectly rank-order every experiment in terms of the latent truncation parameter. Additionally, we validate the RND score by verifying that it measures signal, rather than noise, in Appendix A.4. More details about the training setup and models are in Appendix A.2, A.3.

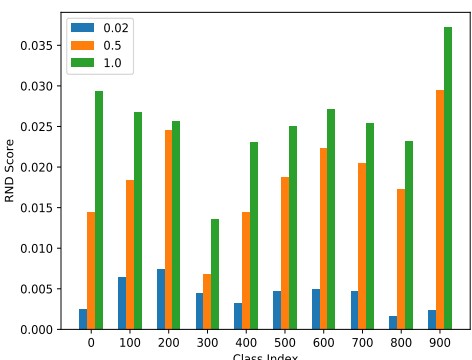 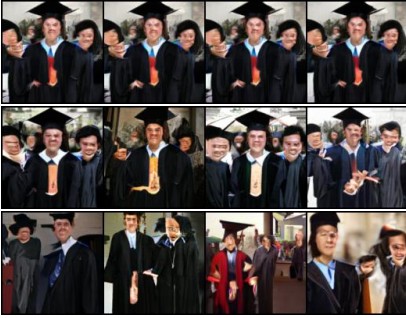

Figure 1: **Left:** RND scores (↑) for BigGAN images generated from latent distributions with various truncations. **Right top:** four images from the least diverse (most severely truncated) distribution (0.02). **Right middle:** four images from the second least diverse distribution (0.5). **Right bottom:** four images from the most diverse distribution (1.0). All images randomly selected from generated data from the ImageNet class "Academic Gown" (index 400).

We include baseline comparisons to both intra-class FID, and the recall metric proposed in Kynkäänniemi et al. (2019) in Table 1. We find that for these truncation experiments, the RND score matches both the human perceived increase in diversity, and the diversity measured by well established metrics.

### 4.2 IMAGENET GANS

ImageNet (ILSVRC2012 dataset - Russakovsky et al. (2015)) is widely considered a difficult task for generative models because of its size and diversity (Brock et al., 2018). Fittingly, we benchmark

Table 1: Baseline comparisons to other metrics. **Left:** intra-class FID (↓) scores on the truncation experiment detailed in Fig. 1. **Right:** Recall (k=5) (↑) scores on the same truncation experiment.

| Class idx | 0.02 | 0.5 | 1.0 | Class idx | 0.02 | 0.5 | 1.0 |
|---|---|---|---|---|---|---|---|
| 0 | 66.6105 | 44.5934 | 29.6121 | 0 | 0.0 | 0.5288 | 0.6337 |
| 100 | 42.7727 | 20.1884 | 13.7058 | 100 | 0.3461 | 0.6206 | 0.6323 |
| 200 | 145.0549 | 58.5112 | 30.8629 | 200 | 0.0 | 0.6093 | 0.6259 |
| 300 | 59.4561 | 28.2421 | 14.2526 | 300 | 0.0209 | 0.6201 | 0.6337 |
| 400 | 134.0554 | 87.4824 | 40.7599 | 400 | 0.0517 | 0.6064 | 0.6196 |
| 500 | 70.3494 | 45.6860 | 29.2421 | 500 | 0.1909 | 0.5795 | 0.6254 |
| 600 | 232.9096 | 92.5253 | 64.2264 | 600 | 0.2182 | 0.5986 | 0.5991 |
| 700 | 236.5844 | 122.7225 | 68.5625 | 700 | 0.0 | 0.6235 | 0.6245 |
| 800 | 136.2272 | 83.9329 | 57.2190 | 800 | 0.0 | 0.5322 | 0.6318 |
| 900 | 78.18263 | 49.4285 | 33.6281 | 900 | 0.0 | 0.5927 | 0.6015 |

well-known generative models of varying recency on several ImageNet classes. In our comparison, we include BigGAN ($128 \times 128$), Self-Attention GAN (SAGAN), and AC-GAN (Brock et al., 2018; Zhang et al., 2019; Odena et al., 2017). We find that the newer generative models, BigGAN and SAGAN, generally produce more diverse images than the older AC-GAN model. However, our metric is able to detect a class where the SAGAN model collapses and produces homogeneous images (see Figure 2). Note that the training procedure in calculating RND scores varies from that of the truncation experiments due to the differences in settings such as image size. More details on this experiment can be found in appendix section A.3.

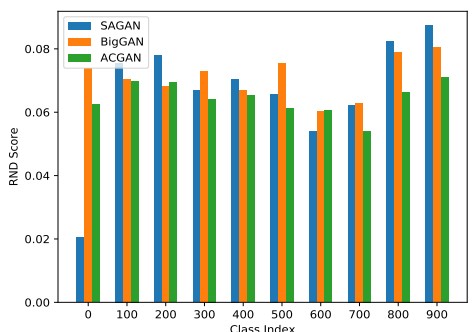 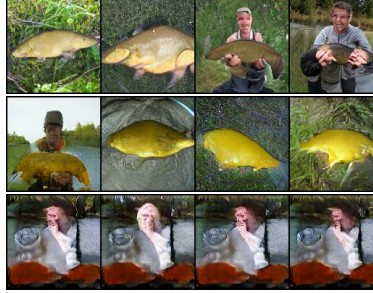

Figure 2: **Left:** RND scores (↑) for different ImageNet $128 \times 128$ GANs. **Right top:** four images from BigGAN. **Right middle:** four images from ACGAN. **Right bottom:** four images from SAGAN. All images randomly selected from generated data from the ImageNet class "Tench" (index 0). The RND score emphasizes the poor diversity in this class of SAGAN data.

### 4.3 IMAGENET CLASSES

Because our metric does not rely on comparing generated samples to ground-truth data, unlike FID, we are able to compare the diversity of natural data across classes on ImageNet. We compare diversity across the 10 ImageNet classes on which we measured the diversity of ImageNet generators. We present these results in Figure 3. We find that there is a wide range of diversity among ImageNet classes. RND scores for the classes, along with randomly selected example images from the least and most diverse classes are included in Figure 3.

### 4.4 SINGLE-IMAGE GANS

Popular generative models often leverage large amounts of data during training to generate high quality and diverse images. However, a growing body of work aims to generate images when training data is scarce. Recent work has demonstrated unconditional image generation learned from a single image (Shaham et al., 2019a; Hinz et al., 2020). These models leverage the idea that images can be decomposed into patches which can be re-configured to distribute information to parts of a

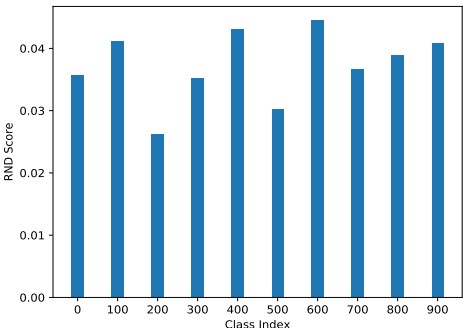
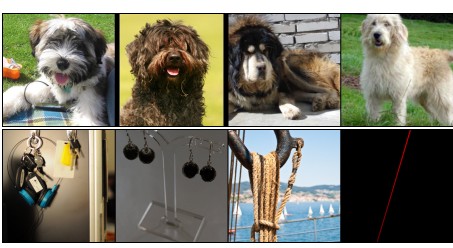

Figure 3: **Left:** RND scores (↑) for ten different ImageNet classes. We observe a range of diversity measurements and display images from the most and least diverse classes. **Right top:** four images taken from the least diverse ImageNet class measured ("Tibetan terrier" - index 200). **Right bottom:** four images taken from the most diverse class measured ("Hook/Claw" - index 600).

synthetic image where it did not exist in the single training sample. In this domain, practitioners hope the generated data will be *more* diverse than the ground-truth data. FID fails to accurately measure diversity in this setting because these methods do not seek to replicate the distribution of data they were trained on. Additionally, in the data-scarce setting, underlying distributions may be very different than the distribution of ImageNet data used to train the feature extractor leveraged in FID. Since our metric does not depend on a ground-truth dataset, we avoid these problems and are able to measure the diversity of images generated by few-shot generative models.

We benchmark two recent methods in the data-scarce regime: SinGAN (Shaham et al., 2019a) and ConSinGAN (Hinz et al., 2020). The authors of SinGAN propose Single-Image FID (SIFID) to measure performance in this domain. SIFID calculates the Fréchet distance between Gaussian distributions fit to early layer features (compared with last layer features of the extractor used by FID). Thus, SIFID also suffers from the problem that simply reproducing the training image minimizes the score. Furthermore, because only one image is used in the feature extraction, SIFID was later found to exhibit very high variance across generated images and does not provide a valuable distinction between "better" and "worse" images (Hinz et al., 2020).

To benchmark these models, we generate a variety of images from randomly selected fixed training images, and we compare the diversity measurements. Results of the experiments can be found in Figure 4. While ConSinGAN claims significant advantages in training time over SinGAN, we find that in many cases, SinGAN actually produces more diverse data than ConSinGAN, confirming the suspicions of Hinz et al. (2020). More details can be found in Appendix A.3.

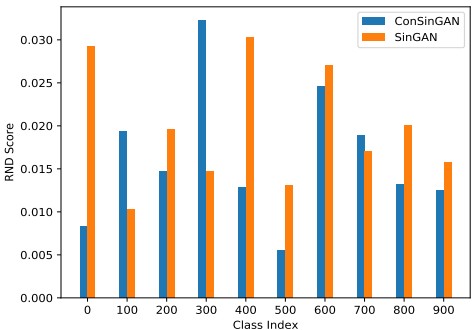
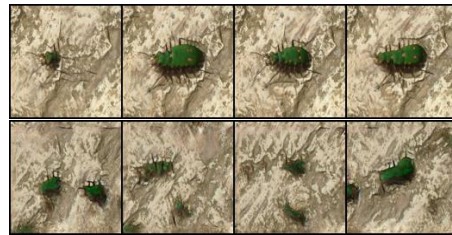

Figure 4: **Left:** RND scores (↑) comparing single-image GANs. **Right top:** four randomly selected images generated from a single ImageNet training image by SinGAN. **Right bottom:** four randomly selected images generated from a single ImageNet training image by ConSinGAN. Images are from the class "Tiger Beetle" (index 300).

## 4.5 CELEBA

In addition to measuring the diversity of a variety of generative models on ImageNet data, we also benchmark several generative models, including a VAE, on unconditional (w.r.t face attributes) CelebA $128 \times 128$ images. The CelebA dataset contains over 200K images of celebrities with different facial attributes (Liu et al., 2018). We measure the diversity of faces generated by RealnessGAN, Adversarial Latent Autoencoder (ALAE), and Progressive Growing GAN (PGAN) (Xiangli et al., 2020; Pidhorskyi et al., 2020; Karras et al., 2017). Our results indicate that newer generative models, like RealnessGAN, outperform the older PGAN model, which tends to produce faces in the same pose and scale, on diversity of generated samples (see Figure 5).

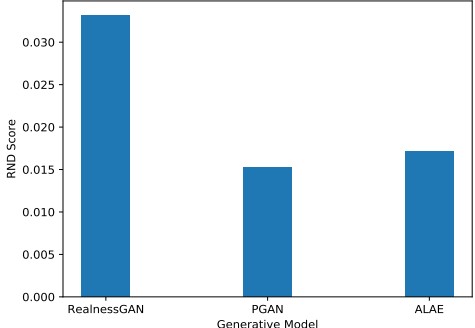 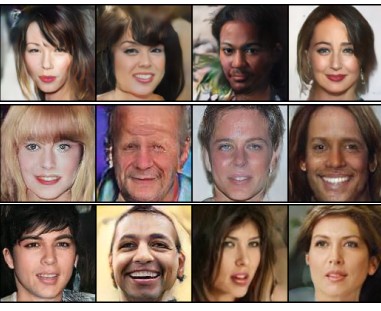

Figure 5: **Left:** RND scores ($\uparrow$) for different CelebA $128 \times 128$ generative models. **Right top:** four images randomly taken from the most diverse data generated by RealnessGAN. **Right middle:** four random images generated by PGAN. **Right bottom:** four random images generated by ALAE.

## 4.6 RND IN A NATURAL LANGUAGE PROCESSING SETTING

In addition to measuring the diversity of generative models for images, we also benchmark several natural language processing models. We seek to answer the following questions empirically: 1. How does RND measure diversity in a controlled setting where we manipulate the size of the model vocabulary? 2. How does RND rank diversity for naturally and synthetically generated text from models of varying capacity?

To answer these questions, we conduct experiments on the WikiText dataset (Merity et al., 2016). The WikiText language modeling dataset is a collection of over 100 million tokens extracted from the set of verified Good and Featured articles on Wikipedia. Following Xu et al. (2018), we use the number of distinct tokens in the vocabulary as a surrogate for controlling diversity. We synthetically manipulate the diversity of the text by truncating the size of the vocabulary. Similar to our experiments in subsection 4.1, this "truncation trick" allows us to improve the text quality at the cost of diversity. We validate our diversity measurement by evaluating text from the WikiText dataset using five different vocabulary sizes: $5k, 10k, 15k, 20k, 25k$, where we keep track of the top-$k$ tokens in the text, and replace the least frequent tokens with an out-of-vocabulary $\langle$unknown$\rangle$ token.

Additionally, we compare the natural text from WikiText to synthetic text from the OpenAI GPT-1 (Radford et al., 2018) and GPT-2 (Radford et al., 2019) models (see Appendix A.3 for details).

We use a Tranformer model (Vaswani et al., 2017) implemented in Pytorch (Paszke et al., 2019) as the model architecture for the predictor and target networks. We use the CMU Book Summary Dataset (Bamman & Smith, 2013) to generate input prompts for the GPT models.

The results are presented in Figure 6, along with RND scores in Appendix Table 4. We plot the generalization gap versus the number of training epochs. The figure shows the results for the controlled experiment where we vary the size of the vocabulary. Here, we see that RND captures the correct ranking of diversity with larger vocabulary sizes achieving higher scores than less diverse text with smaller vocabulary. Also included in the figure are diversity comparisons for the natural versus synthetically generated text. We observe that RND captures the correct ranking of diversity with the more diverse natural text scoring higher than the synthetic text and the more diverse generations from the stronger GPT-2 model scoring higher than generations from the older GPT-1 model.

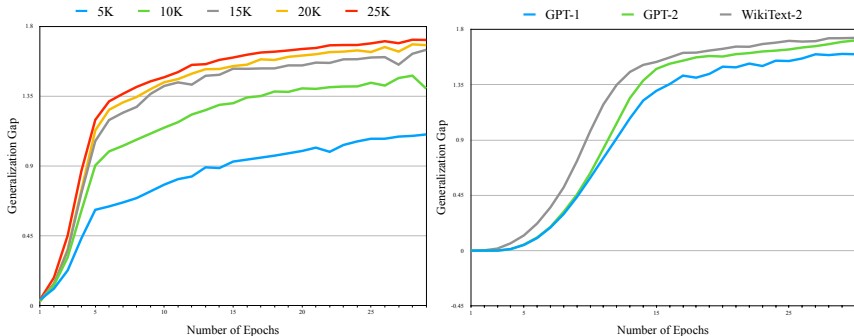

Figure 6: **Left**: Generalization gap for different vocabulary sizes. RND captures the correct ranking of diversity with larger vocabulary sizes achieving higher scores than less diverse, smaller vocabulary text. **Right:** Generalization gap for natural and synthetic text. RND scores the more diverse natural text higher than the synthetic text, with the stronger GPT-2 scoring higher than GPT-1.

## 5 DISCUSSION

We present the need for a diversity metric and introduce a novel framework for comparing diversity based on a technique from reinforcement learning - Random Network Distillation. Our framework avoids many of the pitfalls of existing metrics when it comes to measuring diversity. Furthermore, our metric is flexible and dataset-agnostic, allowing us to measure diversity in several settings including natural images and text, along with a setting for which current metrics are poorly suited: few-shot image generation. We validate our method in a number of controlled scenarios and present results on a wide variety of well-known generative models.

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

# A APPENDIX

## A.1 ALGORITHM

---

**Algorithm 1:** Random Network Distillation Score

---

**Require:** Dataset $S$, training set size $k$, number of runs $r$, start and end RND score epochs, $n_0$
$\quad\quad\quad$ and $n$, and network architecture $\mathcal{A}$.
**for** $j$ *in* $\{1, \ldots, r\}$ **do**
$\quad$ Randomly initialize target network $\mathcal{T}$ and predictor network $\mathcal{P}$.
$\quad$ Randomly split $S$ into training set $S_t$ with $k$ elements, and validation set $S_v$ ;
$\quad$ **for** $i$ *in* $1, \ldots, n$ **do**
$\quad\quad$ Calculate $RND_i$ as in eq. 2
$\quad\quad$ Update parameters of $\mathcal{P}$ to minimize eq. 1
$\quad$ **end**
$\quad$ Calculate $\widehat{RND}^j = \sum_{i=n-n_0}^{n} RND_i$, the average $RND$ value for run $j$.
**end**
**Return:** $\frac{1}{r} \sum_{j=1}^{r} \widehat{RND}^j$

---

## A.2 GENERATOR DETAILS

***ImageNet*** For validating our method on BigGAN truncation data, we generate images from a given class at resolution $256 \times 256$ using pretrained weighs from the BigGAN TF Hub at a truncation value of $1.0$ unless otherwise specified. (`https://colab.research.google.com/github/tensorflow/hub/blob/master/examples/colab/biggan_generation_with_tf_hub.ipynb`). For comparing ImageNet generative models, we generate images at the $128 \times 128$ resolution since this is the resolution at which a majority of the models we test produce images. We use pretrained SAGAN, and ACGAN models taken from `https://github.com/ilyakava/gan`. The SAGAN baseline model achieves an FID score of $16.39$. The ACGAN baseline model achieve an FID score of $24.72$. For the BigGAN generated images, we again use pretrained weights from the BigGAN TF Hub. Details about the FID scores of BigGAN can be found in Brock et al. (2018). For $128 \times 128$ GAN comparisons, we use $100$ training points as compared with $200$ for the full resolution, BigGAN benchmark

***Single-Image GANs*** To test single-image GANs, we fix randomly selected base image IDs for each ImageNet class tested, and then generate $200$ images from each base image. Then, we collect all the generated images from multiple ground-truth images, and randomly shuffle these into a training and validation set. All code taken from the respective github pages for Shaham et al. (2019b); Hinz et al. (2020). We perform runs with a training split of $100$ images because of the relative lack of diversity of single-image GANs compared with classical GANs trained on a large amount of data. Experiments were averaged over $20$ runs, on a ResNet18 with final $10$ epochs averaged out of $40$ training epochs.

***CelebA*** We use pretrained models for each of the CelebA experiments. Pretrained models for ALAE, RealnessGAN can be found on their respective github pages. We use the pretrained PGAN model found on the pytorch GAN zoo github `https://github.com/facebookresearch/pytorch_GAN_zoo`. ALAE requires a base image to generate data, so we randomly sample (unconditionally) images from CelebA to generate samples from ALAE.

***NLP:*** As described in Radford et al. (2018), the GPT-1 model consists of a 12-layer decoder transformer with 12 attention heads and $3,072$-dimensional hidden states. GPT-2 (a successor to GPT-1), was trained to predict the next word in 40GB of Internet text. GPT-2 is a larger transformer-based language model trained on a dataset of 8 million web pages.

For the experiments in Figure 6, we use word representations of size $400$, feedforward layers with inner dimensions $400$, multi-head attention with $4$ attention heads. The model is optimized with Adam (Kingma & Ba, 2014) using a learning rate of $0.001$. The learning rate was selected using a grid search on an independent subset of the WikiText-2 dataset using a grid search with value

ranging from $0.01$ to $1e$-7. We use $4$ transformer encoder layers, and we use a simple linear decoder for computing the scores over vocabulary tokens.

## A.3 EXPERIMENT DETAILS

Unless otherwise stated, we run experiments with the following hyperparameter choices:

- *Net*: ResNet-18
- *Learning rate*: 0.01
- *Epochs to average score*: 10
- *Epochs*: 50
- *Number of runs*: 40
- *Training number*: 200
- *Optimizer*: SGD with momentum = 0.9

We perform ablation studies on these choices in appendix A.4. It is worth noting that within a fixed comparison like comparing ImageNet GANs in Figure 2, diversity is can be compared, but comparing diversity scores across experiments is difficult because of changes in the setting such as image size, training/validation split, architecture. We show stability of these choices within a given experiment in appendix A.4, but the choices do affect comparisons across experimental settings.

*Normalization:* To improve comparability across different models, we normalize data to have mean=0, std=1 for each channel before training the predictor network. We estimate the mean and std of the distribution using the generated samples.

*Measurement time:* The measurement time varies depending on how much training data is used, and how many runs the measurement is averaged over. For $20$ runs on an ImageNet class at full resolution, with the predictor net trained for $50$ epochs, the RND score takes approximately $3.85$ gpu-hours to complete on a NVIDIA GV100.

*Hardware:* Experiments were run on a hetergeneous mixture of hardware consisting of NVIDIA GV100 and NVIDIA RTX 2080 Ti gpus.

## A.4 ADDITIONAL EXPERIMENTS

*Does RND measure signal vs. noise?* A good diversity measurement should be able to distinguish between meaningful features in data, and random noise. As RND was able to distinguish these in the setting of RL, we expect the RND diversity measurement to be able to distinguish these as well. To test this, we generate a fixed set of 3-channel noise data (uniform between $[0, 1]$), and compare the RND score to a randomly selected ImageNet class (see Figure 7).. We find natural images are far more diverse than random noise. Averaged over $40$ runs.

*Epochs:* The RND score is calculated by training a predictor net for $n$ epochs, and averaging over the last $n_0$ epochs. We perform ablation studies on the number of epochs the predictor net is trained in Figure 8. We find that while the scale of the RND scores changes slightly, the ordering of the diversity scores does not change.

Additionally, we test the effect of the choice of $n_0$ on the RND score in Figure 9. We find there is leeway in this choice of hyperparameter as averaging over 10, 15, 20 epochs all produces the same ordering in truncation diversity experiments.

*Number of Runs:* Each run in the calculation of RND corresponds to a new instance of randomly initialized target and predictor networks. We test the effects of the number of runs used in calculating the RND score, finding that the relative ordering of diversity for a given class is preserved (see Figure 10).

*Confidence Intervals:* For the truncation studies in Figure 1, we perform 50 runs resulting in confidence intervals given in Tables 2, 3. The user may tune the number of runs as a hyperparameter with the tradeoff between time and confidence interval width.

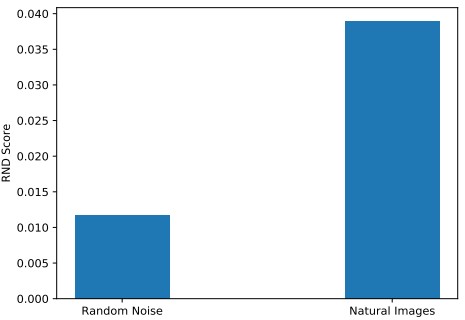 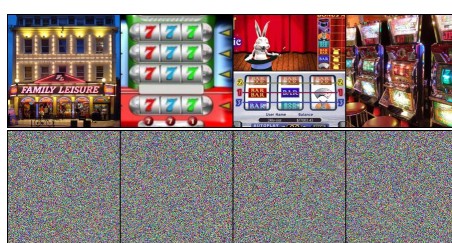

Figure 7: **Left:** RND scores (↑) for random noise, and natural images taken from a randomly selected ImageNet class. **Right top:** four images taken from the ImageNet class measured ("Slot" - index 800). **Right bottom:** four images of random noise used to calculate RND score for random noise.

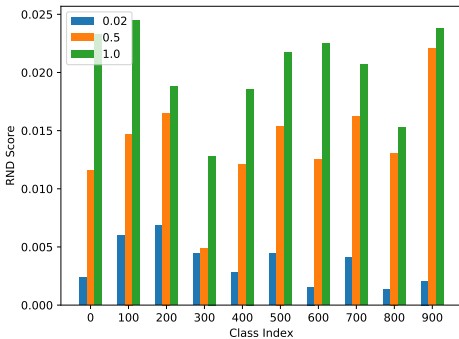 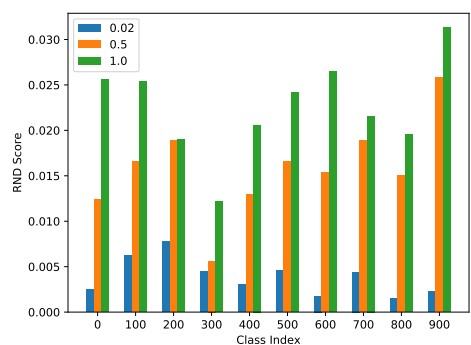

Figure 8: Ablation on epochs used to train the predictor net. Truncation experiments repeated as in 1. **Left:** predictor net trained for 30 epochs. **Right:** predictor net trained for 40 epochs. While we see variation in the scores, the relative ordering is preserved across different networks.

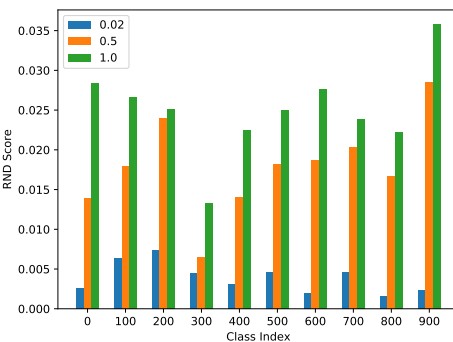 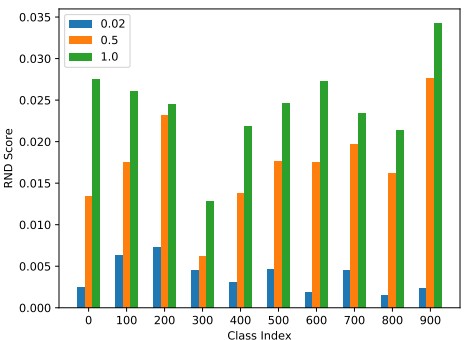

Figure 9: Ablation on epochs used to average in calculating RND score. Truncation experiments repeated as in 1. **Left:** RND calculated with final 15 epochs. **Right:** RND calculated with final 20 epochs. The relative ordering is preserved across different networks.

***Training/Val split:*** A hyperparameter choice we make when calculating the RND score is the number of datapoints in the training set used for distillation. Heuristically, the amount of training data will make a difference in the RND score in the limiting cases. Too few training data, and the

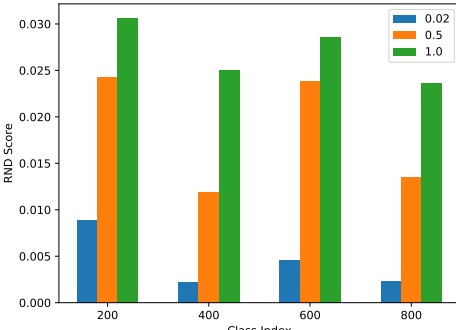 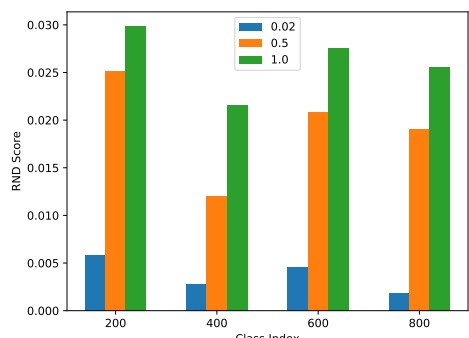

Figure 10: Ablation on runs used to calculate RND score. Truncation experiments repeated as in 1. **Left:** 30 runs used in the calculation. **Right:** 40 runs used in the calculation. The relative ordering is preserved across number of runs.

Table 2: RND ($\uparrow$) scores with confidence intervals for different truncation parameters.

| Class idx | 0.02 | 0.5 | 1.0 |
|:---:|:---:|:---:|:---:|
| 0 | $0.0025 \pm 0.0011$ | $0.0144 \pm 0.0059$ | $0.0293 \pm 0.0053$ |
| 100 | $0.0064 \pm 0.0015$ | $0.0184 \pm 0.0059$ | $0.0268 \pm 0.0056$ |
| 200 | $0.0074 \pm 0.0016$ | $0.0246 \pm 0.0063$ | $0.0257 \pm 0.0044$ |
| 300 | $0.0045 \pm 0.0008$ | $0.0068 \pm 0.0065$ | $0.0136 \pm 0.0061$ |
| 400 | $0.0032 \pm 0.0011$ | $0.0145 \pm 0.0059$ | $0.0231 \pm 0.0073$ |
| 500 | $0.0047 \pm 0.0013$ | $0.0187 \pm 0.0074$ | $0.0250 \pm 0.0083$ |
| 600 | $0.0049 \pm 0.0017$ | $0.0223 \pm 0.0098$ | $0.0271 \pm 0.0097$ |
| 700 | $0.0047 \pm 0.0016$ | $0.0205 \pm 0.0081$ | $0.0254 \pm 0.0061$ |
| 800 | $0.0016 \pm 0.0007$ | $0.0173 \pm 0.0036$ | $0.0232 \pm 0.0072$ |
| 900 | $0.0024 \pm 0.0012$ | $0.0295 \pm 0.0083$ | $0.0372 \pm 0.0120$ |

generalization gap will be large for any dataset, diverse or not. Too many training data, and the generalization gap for diverse sets will be small provided the network accurately learns the features of the target network. This is because the distribution will be "saturated" with training data, and the predictor network will be able to perform well on unseen data. However, we find that the relative ordering of diversity is stable for a wide band of training splits (see Figure 11).

*Network Architecture:*

Table 3: RND ($\uparrow$) scores for different ImageNet classes.

| Class idx | RND score |
|:---:|:---:|
| 0 | $0.0358 \pm 0.0135$ |
| 100 | $0.0411 \pm 0.0134$ |
| 200 | $0.0263 \pm 0.0072$ |
| 300 | $0.0352 \pm 0.0110$ |
| 400 | $0.0431 \pm 0.0124$ |
| 500 | $0.0302 \pm 0.0076$ |
| 600 | $0.0445 \pm 0.0126$ |
| 700 | $0.0367 \pm 0.0114$ |
| 800 | $0.0389 \pm 0.0088$ |
| 900 | $0.0409 \pm 0.0106$ |

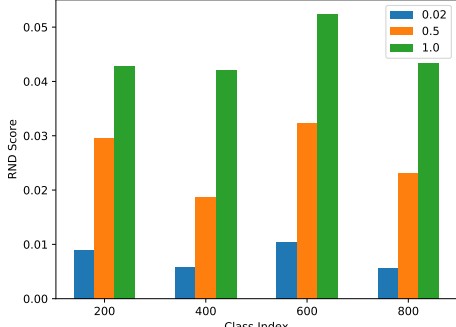 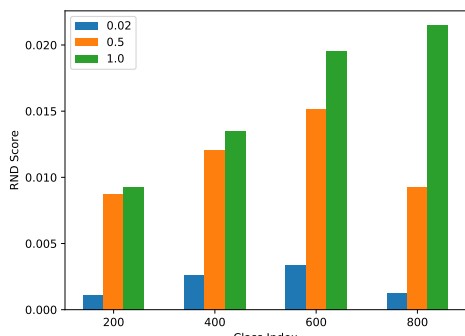

Figure 11: Ablation on training split. Truncation experiments repeated for selected indices as in 1. **Left:** training data set to 100. **Right:** training data set to 500. While we see variation in the scale of the scores, the relative ordering is preserved across different training splits.

We test the sensitivity of the RND score to a change in network architecture. Results of these experiments are presented in Figure 12. We find that there is change in the RND score across architecture, but the relative ordering of datasets of known variation in diversity remains unchanged. Changes in architecture may be necessary when dealing with more or less complex data, as well as settings such as text. Because of this, we stress that RND is best viewed as an intra-setting diversity measurement because of this.

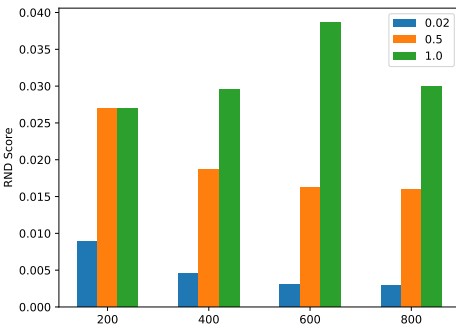 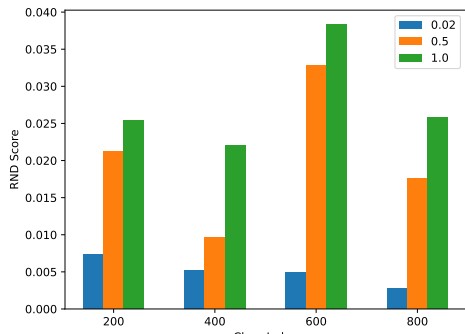

Figure 12: Ablation on network used to calculate RND score. Truncation experiments repeated for selected indices as in 1. **Left:** Alexnet results (Krizhevsky et al., 2012). **Right:** ResNet34 Results. While we see variation in the scores, the relative ordering is preserved across different networks.

***NLP RND Scores:*** The numerical RND scores for the experiments described in Figure 6 are presented in Table 4.

Table 4: RND Scores for NLP. **Left:** Validating RND on NLP with different vocabulary sizes. **Right:** Comparing different generated texts to natural text.

| Vocabulary Size | RND Score |
|---|---|
| $5k$ | 1.0537 |
| $10k$ | 1.4256 |
| $15k$ | 1.5877 |
| $20k$ | 1.6446 |
| $25k$ | **1.6860** |

| Model | RND Score |
|---|---|
| GPT-1 | 1.5561 |
| GPT-2 | 1.6439 |
| WikiText-2 | **1.6947** |

## A.5  VISUALIATION

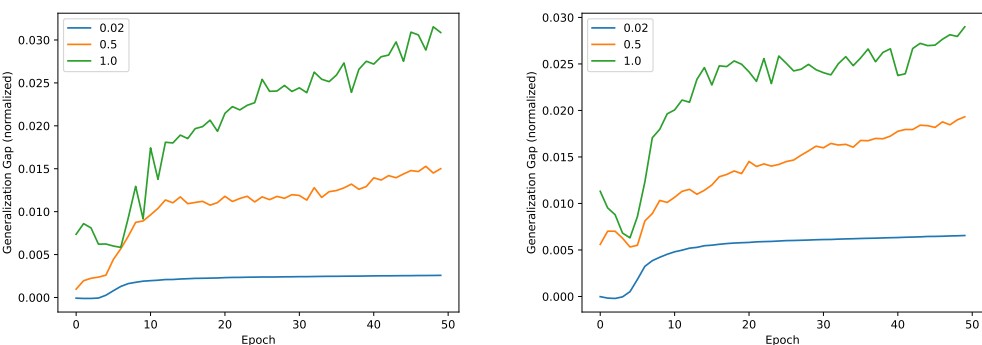

Figure 13: Visualizing generalization gap during training. **Left:** Index 0. **Right:** Index 100

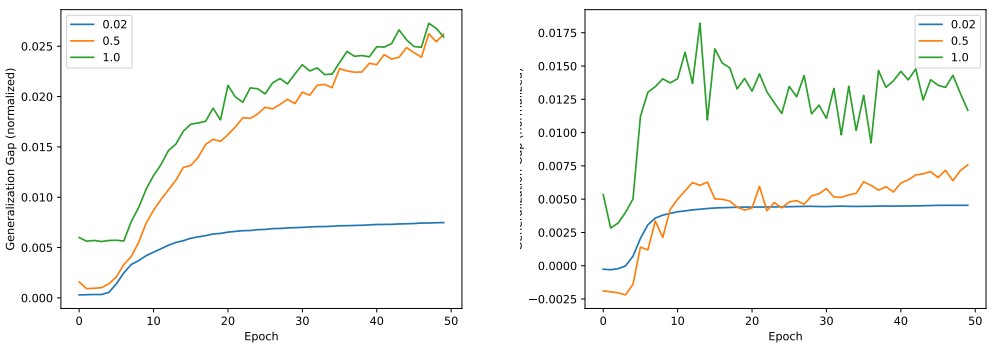

Figure 14: Visualizing generalization gap during training. **Left:** Index 200. **Right:** Index 300

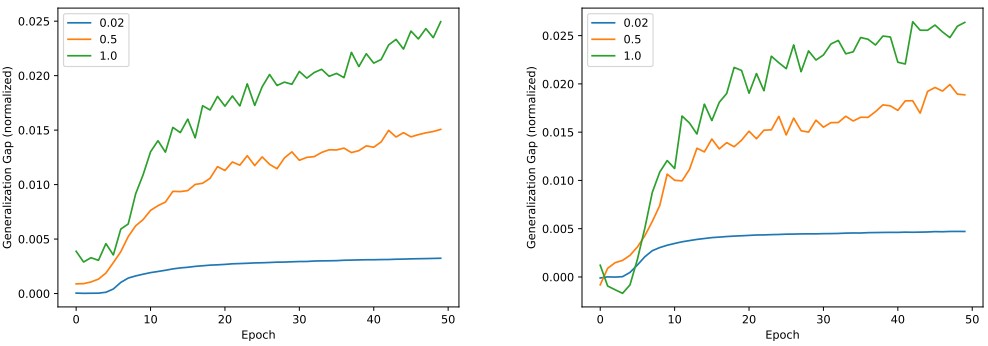

Figure 15: Visualizing generalization gap during training. **Left:** Index 400. **Right:** Index 500

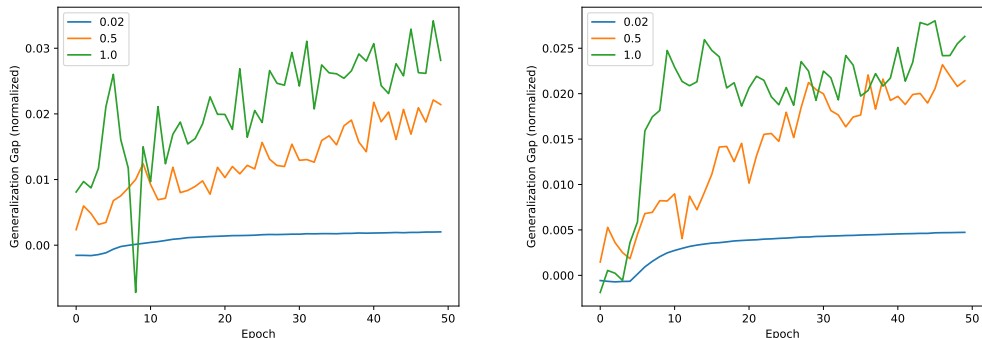

Figure 16: Visualizing generalization gap during training. **Left:** Index 600. **Right:** Index 700

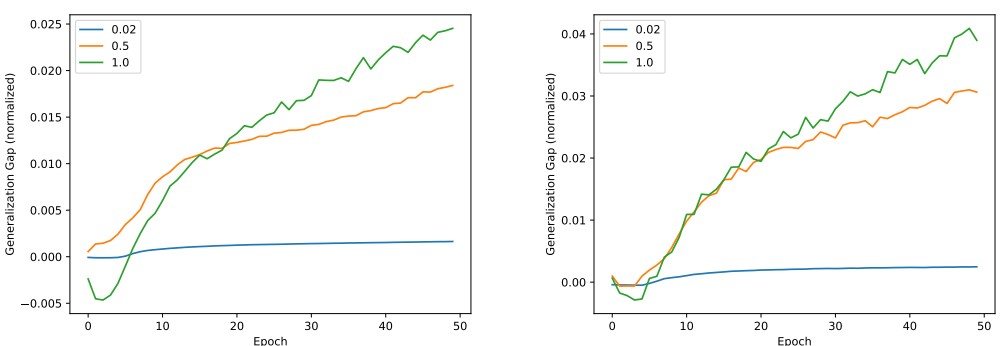

Figure 17: Visualizing generalization gap during training. **Left:** Index 800. **Right:** Index 900

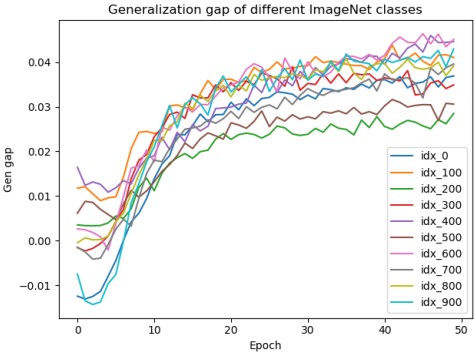

Figure 18: Generalization gap during training on 10 ImageNet classes.

