# OpenReview forum: "Random Network Distillation as a Diversity Metric for Both Image and Text Generation"
_ICLR.cc/2021/Conference — Reject_

### Official Review · AnonReviewer3 · 2020-10-27
**Interesting read, but not ready for publication**

**Rating:** 4
**Confidence:** 4

**Review:**

In this paper, the authors introduce a new quantitative diversity measure advocating its usage for generative models evaluation. In a nutshell, to measure the diversity of a particular set, the authors split it into disjoint train/val subsets and learn a DNN to predict the outputs of another randomly initialized DNN on the train set. Then the generalization gap of the trained DNN is computed on the unseen val subset, and the normalized value of this gap (averaged over several splits/initializations) is considered as a diversity measure.

Pros:

(1) The authors tackle an important problem since the established measures like FID are known to sacrifice diversity in favor of perceptual quality.

(2) The proposed measure is novel, the usage of random networks in a new context sounds interesting.

Cons:

(1) The authors do not relate their measure to the very relevant line of existing works on measuring diversity via Precision/Recall:

Assessing Generative Models via Precision and Recall. NeurIPS 2018

Improved Precision and Recall Metric for Assessing Generative Models. NeurIPS 2019

Reliable Fidelity and Diversity Metrics for Generative Models, ICML 2020

Without explicit highlighting of RND's advantages over the Recall/Coverage measures, I cannot recommend to accept the paper.

(2) The computation of RND requires several DNN trainings, which is time-consuming. This makes RND inconvenient for broad usage, and almost impossible to use in day-to-day research, e.g., for monitoring the training progress.

(3) I am not quite convinced by the experiments, which support the RND applicability. For me, the most sensible experiment is in section 4.1, which shows that aggressive truncations decrease RND. Sections 4.2/4.5 show that newer GAN models typically achieve higher RND compared to older ones, but I cannot consider this as strong evidence, since we do not know if the advantage of newer models comes from diversity rather than perceptual quality.

Overall, my current recommendation is (4), mostly because of missing a crucial part of related work and unconvincing experiments.

::::::Post-Rebuttal update::::::

After reading the new revision, I decided to keep my initial score. I do not consider the need of groundtruth real data for metric computation as a strong disadvantage. The authors report some numbers on Recall in Table 1 but it only shows that Recall is consistent with RND, being much cheaper to compute. Therefore, I do not see any reason to prefer RND over established diversity metrics.

---

> ### Author Response · Authors · 2020-11-24
> **Response to Reviewer 3**
>
>
> 1) Regarding references to related work: we focus on FID because of its popularity. We also do not include comparisons to a number of other proposed diversity metrics precisely because a selling point for our metric is it can be calculated in settings where other metrics, like FID, cannot. However, for our truncation experiments, we agree that it would be helpful (for validation) to have comparisons to other established diversity metrics, and thus we have included measurement numbers and citations for intra-class FID, and improved recall (see general comments). Finally, we would like to note that our criticisms of FID apply to the metrics you cite as they all essentially aim to estimate likelihood of real data under the generated distribution, and thus require access to ground truth data, and are maximally diverse when a generative model simply memorizes the training data.
>
> 2) Regarding the computation time of RND: One can speed up RND by decreasing the number of runs, which will in turn increase the variance of the score. We would also like to note that calculating intra-class FID on a large dataset like ImageNet is also quite time consuming, and is often estimated with smaller sample sizes.
>
> 3) Regarding the convincingness of our experiments: we agree diversity is independent of perceptual quality. However, we tease apart the effects of diversity on the RND score in the truncation experiments, where RND matches the known human evaluation of increased diversity, as well as in the noise experiments in Appendix A4.

---

### Official Review · AnonReviewer4 · 2020-10-28
**An intuitive method for quantifying diversity, but the paper is missing baselines.**

**Rating:** 5
**Confidence:** 4

**Review:**

This paper applies random network distillation (RND) as a method for quantifying how diverse samples from a generative model are. Samples from the generative model (or any dataset) are used to train a neural network to mimic a randomly initialized network. Intuitively, this is a more difficult task on a more diverse dataset, and so the distillation loss can be interpreted as a measure of diversity. The authors argue that this approach has advantages over other diversity metrics because it can capture semantic diversity and does not require a second reference dataset.

Strengths of paper:
+ This article is well-written. The motivation and approach are very clear.
+ The technique is demonstrated in several different domains, including image generation, text generation, and one-shot GANs.
+ The approach is intuitive and agrees with qualitative notions of diversity across each domain it was tested in.

Weaknesses of paper:
- The original contribution is minimal as RND distillations loss is a known technique for quantifying exploration. The main originality comes from identifying it as a way to also quantify diversity in generative dataset.
- The distillation loss metric is not compared to other diversity metrics. This would help demonstrate that the RND score is better aligned with diversity than other standard metrics.
-  The claim that the RND score captures semantic diversity is not well supported. This deserves some scrutiny as the RND is a random feature detector, so it is not clear why it will generally favor semantic diversity. There is an experiment in the appendix to show that the RND score was greater for natural images than random ones,  but it is unclear whether other statistics of the random noise were controlled to make this a fair comparison. This should be expanded to determine that it generalizes.

Overall, while the paper has some merits, it needs to compare its metrics to other available ones to better make its argument.

Comment:
The NLP model needs to be initialized with real text. It would be interesting to dive deeper into how context affects the diversity of the generated text.

---

> ### Author Response · Authors · 2020-11-24
> **Response to Reviewer 4**
>
> 1) As to the original contribution of the paper: while RND exists as an RL bonus, we adapt this into a metric by motivating and evaluating the normalized generalization gap between seen and unseen data - a measurement and distinction that are both novel contributions. Furthermore, we make several practical contributions as we are the first to propose and experiment using a stable metric for diversity on few-shot image generation, as well as natural data.
>
> 2) Regarding comparison to other metrics: we do not include baseline comparisons in many of our experiments exactly because a selling point for our metric is it can be calculated in settings where other metrics, like FID, cannot. However, for our truncation experiments, we agree that it would be helpful (for validation) to have comparisons to other established diversity metrics, and thus we have included measurement numbers for intra-class FID, and recall (see general comments).
>
> 3) Regarding The claim that RND captures semantic diversity: we would like to clarify that the images are normalized to have mean 0, std 1 in each channel. For our noise experiments, the noise generated has these same statistics for a controlled comparison.

---

### Official Review · AnonReviewer2 · 2020-10-28
**Not confident on if this paper proposes a good definition on diversity.**

**Rating:** 4
**Confidence:** 3

**Review:**

I understand the authors goal on developing a diversity metric and evaluate models diversity. However, the concept of diversity is related to multiple factors and I don't agree to define diversity independent of memorization: a good memorization of diverse data also contributes to the diversity of models and sometimes good memorization suggests high capacity of model thus may lead to high diversity.
In the approach proposed, the memorization concept is implicitly wrapped into the size of the predictor model.  But there is no analysis on the effect of the predictor model sizes on the diversity scores.
For the experiment section, the evaluations are rather non-systematic, only a few categories' RND scores are shown. A table of overall performance will be good.
Measuring diversity of models is an important task, while this paper provides some interesting discussion on defining the diversity and proposed method to measure it. but the definition needs a bit refinement and the the author failed to prove that the propose method is a systematic metric on the diversity of models (only showed specific categories is not enough).

---

> ### Author Response · Authors · 2020-11-24
> **Response to Reviewer 2**
>
> 1) Concerning the definition of diversity - we agree that memorizing a very diverse dataset produces a diverse output. Our claim was simply that memorizing a dataset doesn’t automatically yield diversity. Diversity should not be upper bounded by memorizing a particular dataset, as is the case when measuring diversity with intra-class FID.  While CIFAR-10 is diverse, and thus a GAN which memorizes CIFAR-10 produces diverse data, this dataset is not maximally diverse, and we want a measure which recognizes when a GAN produces even more diverse data.
>
> 2) We do include an analysis of architecture choice on the RND score in Appendix A4. We find that comparisons of diversity under our metric are stable across such changes. Furthermore, we would like to point out that other well accepted diversity metrics, like FID, also require architecture choices that may affect the measurement.

---

### Official Review · AnonReviewer1 · 2020-11-02
**Review and Questions**

**Rating:** 6
**Confidence:** 3

**Review:**

RND as a Diversity Metric
This paper proposes a new modality-independent diversity measure for generative models and examines this across image and natural language generation.  The idea repurposes an exploration technique from reinforcement learning:  random network distillation. The method produces a diversity score of data by splitting it into train and validation partitions. Then, a predictor network is trained via a mean-square error (MSE) to predict the resulting features of passing the train data through a fixed, randomly initialized target network. The diversity measure is then computed as the normalized MSE difference between the train and validation partitions. The intuition of the method is that if the train partition is diverse, then we would expect the predictor network to generalize well in predicting validation target features. If it’s not diverse, we would expect a large gap.

I recommend acceptance. This appears to be a useful advance as a diversity measure that works across different modalities. I’m not aware of prior work doing this. The largest weakness, IMO, is that the work doesn’t do enough study into the importance and nature of the random target network. I imagine this is a critical decision (e.g. don’t use a MLP for a vision target network) and, if the authors want this to be widely adopted by different communities, should provide further guidance on this.

Notes:
* This has obvious failure modes. For instance, if the target network was a 0-network (all inputs mapped uniformly to a 0-vector), this trivially fails as a diversity measure. This paper should address more details about the requisite nature of the target network. This is the biggest weakness of this paper and I would upgrade my score with a more thorough scientific investigation here.  “The exact architecture and training procedure depends on the setting. For example, we use a transformer architecture to evaluate text, and we use a ResNet architecture to evaluate images (Vaswani et al., 2017; He et al., 2016).”
* I enjoyed Section 2.1 “What do we want from a diversity metric?”. Capturing the notion of diversity, distinct from information-theoretic measures, is an important property.

Questions to authors:
* What is the importance of the random network architecture? How does the “architectural prior” impact the efficacy of your approach as a data diversity measure?
* Could you improve the clarity of Section 3.1? It reads as, “The bonus is tied to how “novel” an agent’s environment is - as measured by the distillation loss between a fixed, random target network, and a predictor network trained to mimic the features of the target network.”  I found myself re-consulting the original paper to make sure I knew what was going on. I found Section 3.2 to be much clearer.

---

> ### Author Response · Authors · 2020-11-24
> **Response to Reviewer 1**
>
> 1) Concerning the “architectural prior” and training procedure, while you are correct that there are obvious failure cases (i.e. when the target network identically maps to 0), our experiments in Appendix A4 suggest that diversity comparisons made using the RND score are stable across architecture and training hyperparameter choices.  Moreover, common initializations empirically seem to avoid potential degeneracies of the target network.  Finally, we would like to note that other measures, such as FID and recall, also require architectural choices.
>
> 2) Thank you for pointing out the lack of clarity in 3.1.  Upon your suggestion, we have reworded this sentence accordingly.

---

### Official Review · AnonReviewer5 · 2020-11-04
**Interesting diversity measure idea, insufficient comparisons to other approaches**

**Rating:** 4
**Confidence:** 3

**Review:**

This paper proposes that generalization performance of distillations of random networks can be used as a good metric for the diversity of a data set: as a data set gets more diverse, it should be harder to learn to mimic a random computation on that data set.  After defining the metric, the paper compares it to FID in its ability to distinguish truncated GAN output, and applies it to compare different generative architectures and training settings, to compare different data sets such as imagenet classes, and to measure diversity of natural language model outputs.

The strength of the paper in its interesting viewpoint, that diversity can be viewed as the difficulty of a random learning task.  The framework and concept is promising.  It is good to see that, unlike FID, it detects the loss of diversity as a generator is truncated, without mixing the measurement with precision. And the comparisons of different models and data sets is interesting.

However, in proposing that RND should be used as a diversity metric, the paper does not sufficiently compare the proposed method to previously proposed alternatives.  The paper should establish that the metric is meaningful and useful.  Three are three main issues.

1. What is being measured by RND?  Beyond just the operational definition of how the metric is collected?  In what situations would measuring this quantity be expected to differ from other metrics, and what strength weaknesses do the proposed metric have compared to other methods?  It seems possible that the idea of diversity-through-learning complexity could have an interesting theoretical definition, but the paper does not attempt a theoretical characterization of exactly what quantity would ideally be estimated by the RND procedure.

2. How does it compare to previously proposed metrics? A comparison to FID is done, but there are many other approaches for measuring diversity.  E.g., see [Borji 2018] for a survey of a large number of alternative metrics for diversity, many of which are designed to measure recall of a generative model compared to a known diverse ground truth.  Or see [Sajjadi 2018] or even the Parzen windows of [Goodfellow 2014].  Since RND measures diversity without need for a ground truth, it could be argued that RND allows new measurements.  But to establish the metric is sound, it should first be compared to a variety of recall metrics where a ground truth is known.  Comparisons should be done with different types of recall measures, and also different types of data distributions including toy examples where differences in diversity are easy to understand.

3. It is a goal for RND to match human assessments of diversity?  It is argued that some complexity (such as noise) is uninteresting and should not be included in a diversity metric - but this seems to imply that the goal is to measure just the diversity that would be interesting or informative to a human.  If this is the goal, then the performance of RND should be compared to human assessments of diversity.

The idea and the topic of the paper is interesting and strong, but needs further development to argue that the proposed measure is meaningful and useful, especially compared to other approaches.

---

> ### Author Response · Authors · 2020-11-24
> **Response to Reviewer 5**
>
>
> 1) Simply put, the aim of RND is to measure the diversity of data. While there does exist theoretical work examining the utility of random features, we instead focus on empirical validation of our method. As for situations in which this metric will differ from other metrics, we would like to stress that the benefit of our metric is that it is more flexible than existing metrics. We do not include comparison/baseline numbers in many of our experiments because other diversity metrics often require access to ground-truth distributions in order to quantify diversity (see more below).
>
> 2) We focus on FID due to its popularity. As for baseline numbers, we did not include them originally because a selling point for our metric is it can be calculated in settings where other metrics, like FID, cannot. However, for our truncation experiments, we agree that it would be helpful (for validation) to have comparisons to other established diversity metrics, and thus we have included measurement numbers for intra-class FID, and recall (see general comments).  Nonetheless, our criticisms of FID apply to all of the metrics you mention, as they all require a ground truth distribution, and essentially seek to measure the likelihood of real data under the generated distribution. Our claim is not that these methods fail in the most obvious settings, but rather they are inflexible. In many of the experiments we run, we do not include a comparison to these existing measurements because they simply cannot be run on settings like few-shot image generation and the diversity of a natural image dataset.
>
> 3) We do wish to match human assessments of diversity with RND, although "interesting diversity" remains somewhat nebulous, which is why we focus on desiderata like robustness to small amounts of noise. As for comparing our metric to human perception, to our knowledge, there do not exist substantial datasets on which human assessment of diversity has been thoroughly measured. We do however match human assessments of diversity in the ImageNet truncation experiments, where RND matches the perceived increase in diversity that comes with a larger truncation parameter.

---

### Author Response · Authors · 2020-11-24
**General Comments**

We thank the reviewers for their feedback. In response to a common concern raised about baseline comparisons to other metrics, we have included two new tables in the submission. These can be found in section 4.1. We would like to stress though that we do not include comparison numbers for many of the experiments we run because many existing metrics cannot be used to measure diversity in few-shot settings, or the diversity of natural data. We will also address each reviewer’s specific points below.

---

### Decision · Program_Chairs · 2021-01-07
**Final Decision**

**Decision:**

Reject

**Comment:**

The paper proposes the generalization performance of distillation from random networks as a metric of diversity, named RND. Intuitively, the more diverse the generated datasets, the more difficult it should be for a model to learn a random computation. The reviewers agree that the metric has a novel perspective. Unfortunately, the paper is not sufficiently developed to be accepted at this point. It is currently missing a number of experiments that would demonstrate that this metric is indeed a measure of diversity:

1.) RND shows sensitivity to the truncation trick in GANs (for images), and limiting the size of vocabulary in text, but does not show sensitivity to any other changes in diversity (such as human judgment of diversity)
2.) It does not compare to previous metrics of diversity, of which there are many
3.) How sensitive is RND to architecture choice.
4.) It is non-obvious to what extent the metric is sensitive to image/text quality

Strong metrics should demonstrate lack of "failure modes", as the utility of a metric is its inability to be gamed. Currently, the paper does not demonstrate this property, though I imagine that more work will help clear up the strengths and weaknesses of the metric.  As a result, I can only recommend rejection.